# Effect of Xanthan Gum, Kappa–Carrageenan, and Guar Gum on the Functional Characteristics of Egg White Liquid and Intermolecular Interaction Mechanism

**DOI:** 10.3390/foods11142119

**Published:** 2022-07-17

**Authors:** Sijia Gong, Xuefeng Shi, Jiangxia Zheng, Ruitong Dai, Junying Li, Guiyun Xu, Xingmin Li

**Affiliations:** 1College of Animal Science and Technology, China Agricultural University, Beijing 100193, China; g1965176512@163.com (S.G.); xuefeng.shi@cau.edu.cn (X.S.); jxzheng@cau.edu.cn (J.Z.); lijunying@cau.edu.cn (J.L.); 2College of Food Science and Nutritional Engineering, China Agricultural University, Beijing 100193, China; dairuitong2022@163.com

**Keywords:** egg white protein, polysaccharide, functional characteristics, intermolecular interaction, aggregation state

## Abstract

This study evaluated the effects of three polysaccharides, xanthan gum (XG), kappa-carrageenan (CA), and guar gum (GG), on the foaming and emulsifying properties of egg white liquid (EWL) and explored the intermolecular interactions and aggregation states in the initial polysaccharide–EWL complex. The results showed that the addition of XG and GG significantly improved the foaming stability of EWL on the one hand, from 66% to 78% and 69%, respectively (*p* < 0.05). On the other hand, the addition of XG and GG significantly improved the foam uniformity and density, and the average foam area decreased from 0.127 to 0.052 and 0.022 mm^2^, respectively (*p* < 0.05). The addition of XG and CA significantly improved the emulsification activity index (from 13.32 to 14.58 and 14.36 m^2^/mg, respectively, *p* < 0.05) and the emulsion stability index (from 50.89 to 53.62 and 52.18 min, respectively, *p* < 0.05), as well as the interfacial protein adsorption at the oil–water interface; it also reduced the creaming index. However, GG negatively affected these indicators. Furthermore, the electrostatic and hydrophobic interactions among molecules in EWL due to XG and the electrostatic, hydrogen bonding, and hydrophobic interactions among molecules in EWL due to CA ultimately led to the irregular aggregation of egg white proteins. Hydrophobic interactions and disulfide bonds between molecules in EWL–containing GG formed filamentous aggregations of egg white proteins. This work reveals that molecules in the polysaccharide–egg white complexes aggregate by interaction forces, which in turn have different effects on the foaming and emulsifying properties of egg white proteins.

## 1. Introduction

Egg white contains 9.7–10.6% protein, including ovalbumin (54%), ovomucoid (11%), ovotransferrin (12%), ovomucin (3.5%), and lysozyme (3.4%) [1]. The foaming property of egg white depends on the surface activity and film formation of these proteins. With the increasing demand for foam food, more manufacturers use bubbles as food ingredients to meet consumer demand. The performance of egg white foam directly affects the competitiveness of egg white in the market. The better the foaming of egg white protein is, the larger the cake volume obtained by the same quality of egg white liquid in the process of preparing cake [2]. The emulsifying properties are also produced by the amphiphilic nature of egg white protein. In an oil–water mixture, the protein is aligned at the oil–water interface and forms a viscoelastic interfacial film to stabilize droplets and make the emulsion even and delicate. Hydrophobic interactions are the key force in determining the adsorption performance of proteins at the oil–water interface [3]. The hydrophobic amino acids of ovalbumin account for approximately 50% of the total amino acids, which are mostly distributed inside of the spherical protein structure, and this inherent structure limits its functional characteristics [4]. Therefore, the utilization rate of eggs can be enhanced by improving the functional properties of egg white protein, which fully reflects its economic benefits.

The complexes formed by proteins and polysaccharides not only possess nutritional and health properties but also provide improved functional characteristics compared to single macromolecules. Common polysaccharides in the food industry include ionic polysaccharides and nonionic polysaccharides. Ionic polysaccharides include XG, CA, and pectin. Nonionic polysaccharides include GG, dextran, and Arabic gum. Among these, XG, CA, and GG have been widely studied in recent years due to their beneficial effects on food characteristics [5,6,7]. The molecular interaction between polysaccharides and proteins can be used to produce foods with diverse structures, appearances, textures, and functions [8]. Studies have shown that soy protein isolate and glucan under neutral conditions could form a protein–glucan graft through the Maillard reaction (glycosylation), which significantly improved the solubility, emulsifying activity index, and emulsifying stability index of soy protein [9]. However, the Maillard reaction process is difficult to control and easily produces harmful substances. Therefore, a noncovalent composite system prepared by physical mixing could provide a solution to this problem. When polysaccharides and proteins are mixed, the conformation of the protein will be changed by electrostatic attraction and hydrogen bonding between polysaccharides and proteins. These forces contribute to the complex state of biological macromolecules and directly affect the functional characteristics of egg white protein.

The objective of the present work was to investigate the effect of three polysaccharides (XG, GG, and CA) on the functional properties of egg white fluid. We also further explored the intermolecular interactions of the initial polysaccharides–egg white complex liquid to explain the molecular mechanism of different types of polysaccharides affecting the functional characteristics of protein solution.

## 2. Materials and Methods

### 2.1. Sample Collection

Eggs were obtained from the National Center for Performance Testing of Poultry (Beijing, China). XG, CA, and GG were purchased from Beijing Tianhong Shengshi Trading Co., Ltd. (Beijing, China).

### 2.2. Sample Preparation

Egg white liquid was collected and homogenized under a magnetic stirrer at a speed of 1200 r/min for 30 min at 25 °C. Then, 0.2% XG, CA, and GG were added to egg white liquid and placed at 4 °C for 24 h until complete hydration. The groups were named EWL (egg white liquid), EWL + XG (adding xanthan gum to egg white liquid), EWL + CA (adding kappa–carrageenan to egg white liquid), and EWL + GG (adding guar gum to egg white liquid).

### 2.3. Foaming Properties

Foaming ability (FA) and foaming stability (FS) were determined according to the method of Zhao et al. [10]. Egg white liquid (25 mL) was diluted four–fold using distilled water and then stirred at 12,000 r/min for 1 min using a high–speed tissue rammer (DS–1, Shanghai Precision Scientific Instrument Co., Ltd., Shanghai, China). The volume of foam (V_1_), total volume (V_0_), and foam volume after 30 min of whipping (V_2_) were recorded. FA and FS were calculated according to Equations (1) and (2), respectively:(1)$$\mathrm{FA}/\%\;=\frac{V_{1}}{V_{0}}\times 100\;$$
(2)$$\mathrm{FS}/\%\;=\frac{V_{2}}{V_{1}}\times 100\;$$

### 2.4. Foam Microstructure

The foam microstructure was determined according to the method of Ye et al. [11]. Egg white liquid (100 mL) was stirred at 12,000 r/min for 1 min using a high–speed tissue rammer, and the volume of the foam was immediately recorded. An appropriate amount of foam was placed on the glass slide to form a thin layer and then placed under an inverted microscope (IXplore, Olympus Corporation, Tokyo, Japan) and magnified 10 times to observe the microstructure of foams.

### 2.5. Emulsifying Properties

Emulsifying activity index (EAI) and emulsifying stability index (ESI) were determined using the methods described by Ercelebi and Ibanoğlu [12] and Yi et al. [13]. First, 12 mL of egg white liquid and 4 mL of soybean oil (Yihai Kerry Food Marketing Co., Ltd., Shanghai, China) were homogenized at 25000 r/min for 2 min using a homogenizer (FLUKO, Shanghai fluke Technology Development Co., Ltd., Shanghai, China) to obtain the emulsion. The emulsion (50 µL) was removed from the bottom of the container and diluted with 5 mL 0.1% sodium dodecyl sulfate solution (Beijing Solarbio Science & Technology Co., Ltd., Beijing, China). The diluent was measured at 500 nm by a UV–Vis spectrophotometer (Evolution 60 s, Delta Science Co., Ltd., Palmdale, CA, USA). After the emulsion was rested for 30 min, another 50 µL was taken from the bottom of the container and diluted according to the methods outlined above. The result of the emulsion formation was defined as EAI, and the result after 30 min was defined as ESI, and EAI and ESI were calculated according to Equations (3) and (4), respectively:(3)$$\mathrm{EAI}(m^{2}/\mathrm{mg})=\frac{2T}{\emptyset C}$$
(4)$$\;T\;=\frac{2.203A}{L}$$
(5)$$\mathrm{ESI}/\min\;=\frac{A_{0}}{A_{0}-A_{30}}\times 30\;$$
where T is the turbidity, A is the absorbance at 500 nm, L is the optical path length (m), φ is the oil volume fraction of the emulsion, and C is the protein concentration before the formation of the emulsion (mg/mL). The result is expressed as the surface area (m^2^) of the protein per unit weight (mg) used in the emulsion.

### 2.6. Creaming Index

Creaming index (CI) was determined by the method of Erçelebi and Ibanoğlu [12]. The emulsion (10 mL) was transferred to the test tube and stored to evaluate the stability of the emulsions. The serum layer height was measured every 30 min, and the total time was 5 h. CI was calculated according to Equation (6):(6)$$\mathrm{CI}/\%\;=\frac{\mathrm{serum}\;\mathrm{layer}\;\mathrm{volume}\;}{\mathrm{total}\;\mathrm{emulsion}\;\mathrm{volume}\;}\times 100\;$$

### 2.7. Interfacial Protein Adsorption

Interfacial protein adsorption was determined according to the method described by Li et al. [8]. The emulsion (2 mL) was put into a tube and centrifuged at 13,300× *g* for 45 min (D3415R, Biotool Swiss AG, Berna, Switzerland). Then the lower clear liquid was removed with a syringe, and the volume measured with a measuring cylinder. The protein concentration of aqueous phase and of liquid egg white were determined by the glycine and bicinchoninic acid protein assay kit (Beijing Solarbio Science & Technology Co., Ltd., Beijing, China). Interfacial protein adsorption was calculated according to Equation (7):(7)$$\mathrm{Interfacial}\;\mathrm{protein}\;\mathrm{adsorption}\;\left(\mathrm{mg}\right)=\frac{M_{I}-M_{f}}{M_{I}}\;$$
where the total content of protein in 2 mL egg white liquid is M_I_ (mg), and the content of protein in the lower clear liquid of emulsion is M_f_ (mg).

### 2.8. Viscosity Determination

Viscosity determination was carried out using the method described by Chang et al. [14]. Two percent NaCl or urea (Yeasen Biotech Co., Ltd., Shanghai, China) was added to each egg white liquid group and then placed on the sensing plate (C plate) of the rheometer (ARES G2, TA Instrument, New Castle, DE, USA) after hydration for 24 h. The plate spacing was 1000 nm. The excess sample at the edge of the plate was removed with a plastic scraper. The shear rate range was 0.01–1000 s^–1^. The relationship between the shear rate and apparent viscosity was determined. The groups were named EWL, EWL + 2% NaCl, EWL + 2% urea, EWL + polysaccharide (XG, CA, GG), EWL + polysaccharide + 2% NaCl, and EWL + polysaccharide + 2% urea.

### 2.9. Surface Hydrophobicity

Surface hydrophobicity was determined using the method described by Deepika et al. [15]. Egg white liquid was diluted in 0.01 mol/L phosphate buffer (pH 7.0) with a protein concentration between 0.01 and 0.2 mg/mL. Then, 4 mL of different diluent were added to 20 μL of 0.008 mol/L 8–phenylamino–1–naphthalene sulfonic acid (Shanghai Macklin Biochemical Industry Park, Shanghai, China) solution (prepared by 0.01 mol/L pH 7.2 phosphate buffer solution). After shaking and standing for 3 min, the fluorescence intensity of the samples was measured using a fluorescence spectrophotometer (FS5, Edinburgh Instruments Ltd., Edinburgh, UK) without and with 8–phenylamino–1–naphthalene sulfonic acid and recorded as FI0 and FI1. The excitation and emission wavelengths were 395 and 475 nm, respectively, and the slit correction was 5 nm. The difference between the two wavelengths was recorded as FI and was used to prepare a protein concentration curve. The surface hydrophobicity of protein was estimated by determining the slope in the initial stage of the curve.

### 2.10. Sulfhydryl Group Analysis

Sulfhydryl group analysis was determined using the method of Chang et al. [14]. To determine the surface sulfhydryl group content, egg white liquid was diluted to 2 mg/mL with tris glycine buffer (0.086 mol/L Tris, 0.09 mol/L glycine, 4 mmol/L ethylene diamine tetraacetic acid (EDTA), pH 8.0, Beijing Solarbio Science & Technology Co., Ltd., Beijing, China). Then, 6 mL of the protein sample was added to 0.02 mL of Ellman’s test solution (0.2 g Ellman’s Reagent (DTNB) was dissolved in 50 mL buffer). The samples were allowed to react for 15 min and then centrifuged at 4000 r/min for 10 min. The absorbance of the supernatant of the samples was measured at 412 nm (SpectraMax i3x, Meigu Molecular Instrument Co., Ltd., Shanghai, China). To determine the free sulfhydryl group content, egg white liquid was prepared with tris glycine buffer (0.086 mol/L Tris, 0.09 mol/L glycine, 4 mmol/L EDTA, 8 mol/L urea, pH 8.0) with a mass concentration of 2 mg/mL. The subsequent steps were the same as those for the determination of the surface sulfhydryl group content. The sulfhydryl content was calculated according to Equation (8):(8)$$\mathrm{sulfhydryl}\;\mathrm{content}\;\left(\mu \mathrm{mol}/g\right)=\frac{73.53\times A412\times D}{C}$$
where 73.53 is obtained from 106/13,600, 13,600 is the molar extinction coefficient of Ellman’s reagent (L/(mol·cm)), A412 is the absorbance value of the sample when DTNB is added, D is the dilution factor, and C is the mass concentration of the sample (mg/mL).

### 2.11. Confocal Microscopy

Confocal microscopy was carried out using the method described by Kazumasa et al. [16]. The lyophilized egg white powder was dissolved in PBS buffer (Shanghai Baili Biotechnology Co., Ltd., Shanghai, China) to obtain an egg white solution with a protein concentration of 2%. The 2% fluorescein isothiocyanate (Sigma-Aldrich Co., Ltd., Shanghai, China) solution was prepared with dimethyl sulfoxide solution (Sinopharm Chemical Reagent Co., Ltd., Shanghai, China). Then, 25 μL fluorescein isothiocyanate was added to 100 mL protein solution and stirred in a magnetic stirrer for 1.5 h. Then, 0.2% polysaccharide was added to the solution, and a small amount of sample was dropped onto the slide. After covering with the coverslips, they were observed with a super−resolution laser scanning confocal microscope at an emission wavelength of 485 nm (A1HD25, Nikon, Tokyo, Japan).

### 2.12. Statistical Analysis

Experiments of each group were conducted in triplicate. All statistical analyses were performed using SPSS 22.0 (Chicago, IL, USA). Results were considered statistically significant at *p* < 0.05 (Duncan’s one−way analysis), and GraphPad Prism 8.0 (GraphPad Software, SanDiego, CA, USA) was used for graphing. The number and average area of bubbles were calculated by ImageJ (National Institutes of Health, Bethesda, MD, USA).

## 3. Results and Discussion

### 3.1. Effects of Polysaccharides on the Foaming Properties of Egg White Liquid

#### 3.1.1. Effect of Polysaccharides on FA and FS of Egg White Liquid

FA reflects the ability to add air to the protein solution, and FS represents foam volume and bubble size stability over time [10]. As shown in Figure 1, compared with the control group (EWL), FA and FS of the EWL + XG group were significantly increased (121% *vs.* 106%, 78% *vs.* 66%, *p* < 0.05), whereas the FA and FS of the EWL + CA group increased with no significant difference (108% *vs.* 106%, 67% *vs.* 66%). Compared with the EWL group, the FA of the EWL + GG group was significantly decreased (92% *vs.* 106%, *p* < 0.05), whereas the FS was significantly increased (69% *vs.* 66%, *p* < 0.05). The results indicate that polysaccharides significantly influence the foaming properties of EWLs.

While stirring EWL, egg white proteins could diffuse from the water phase and adsorb on the air–water interface because of the compatibility between their hydrophobic and gas hydrophobic regions, resulting in rapid conformational changes and rearrangements. Subsequently, a viscoelastic film was formed via intermolecular interactions [10,17]. Adding XG denatured proteins and decreased the structural order of egg white proteins because of the change in the EWL microenvironment. This is more conducive to enhancing the flexibility of diffusion and distribution of proteins at the gas–water interface and facilitates the formation of an adsorption layer, which promotes capturing and formation of bubbles [18]. Therefore, the FA of EWL + XG group was improved. However, the FA of the EWL + GG group was lower than that of the EWL group, possibly due to the reduction in the flexibility of protein molecules at the air–water interface owing to the structure of the guar polymer chain, which is described in a study of intermolecular interactions [19]. Andréa et al. [5] also found that GG negatively affects the FA of proteins.

As time passes, drainage, coalescence, and disproportionation caused the thinning of the interfacial film, and the foams commenced breakage and collapse. Drainage caused a drier foam when the liquid between bubbles seeped out under gravity, leading to liquid film rupturing between adjacent bubbles. Coalescence refers to the merging of the adjacent bubbles. Disproportionation refers to the transfer of gas from smaller to larger bubbles, resulting in the disappearance of smaller bubbles [10]. The polysaccharides were not inclined to be adsorbed to the air–water interface because of their high hydrophilicity; however, they were combined with egg white protein to cause the hydrophilic region of the protein to further protrude into the aqueous phase, which produced strong steric hindrance protection and inhibited disproportionation and coalescence of bubbles, thereby strengthening FS [17]. More importantly, XG and GG could promote intermolecular interactions to enhance the liquid viscosity, slowing the foam film rupture caused by drainage, which also contributed to FS [20]. Therefore, the FS of EWL + XG and EWL + GG groups was improved. Xie et al. [21] and Mott et al. [22] also reported that the XG solution alone did not exhibit any FA. After compounding with soybean and whey protein isolates, FA and FS of protein were improved, and the essential molecular properties of FA and FS were attributed to the polysaccharide–protein complexes.

#### 3.1.2. Effect of Polysaccharides on Foam Structure of Egg White Liquid

The foam volume after full foaming of the EWL with polysaccharides is shown in Figure 2a. Compared to the control (EWL) group, the foam volume of the EWL + XG and EWL + GG decreased significantly (*p* < 0.05), whereas that of the EWL + CA group was not significantly different. The average area and number of foams are shown in Figure 2b and Table 1. The average foam area of EWL + XG and EWL + GG was significantly smaller than that of EWL and EWL + CA groups. The results indicate that the addition of the two polysaccharides (XG and GG) made the foam area smaller and denser and improved the foam structure of the EWL. In addition, the EWL + GG group had the smallest foam area, while the EWL + XG group had better foam area uniformity.

FS is strongly related to foam microstructure area and distribution [23]. Ye et al. [11] and Zhao et al. [10] found that larger foams were more fragile and could easily and rapidly collapse. Previous studies have also shown that small bubbles can resist combination, expansion, or contraction [24]. In our results, the foam area of the EWL + XG and EWL + GG groups were smaller than those of the EWL + CA and EWL groups. Therefore, the EWL + XG and EWL + GG groups could enhance the stability of the bubbles to resist merging, shrinking, or expanding. Moreover, the foam of the EWL + XG group was more compact and uniform than that of the other groups and could withstand lower capillary pressure, consistent with previous studies [25]; consequently, the FA and FS of the EWL + XG group were the highest.

### 3.2. Effect of Polysaccharides on the Emulsifying Property of Egg White Liquid

#### 3.2.1. Effect of Polysaccharides on Emulsifying Activity and Stability Indexes of Egg White Liquid

EAI indicates the ability of a protein to adsorb at the oil–water interface, and ESI reflects the ability of the protein to resist the aggregation and coalescence of emulsion [26]. Compared with the control group (Figure 3), the EAI (from 13.32 to 14.58 and 14.36 m^2^/mg, respectively) and ESI (from 50.89 to 53.62 and 52.18 min, respectively) of the EWL + XG and EWL + CA groups were significantly increased (*p* < 0.05), whereas the EAI (from 13.32 to 12.17 m^2^/mg) and ESI (from 50.89 to 39.47 min) of the EWL + GG group were significantly decreased (*p* < 0.05).

It is necessary to use a large amount of mechanical energy provided by the homogenization process to increase the interface area and to disperse oil droplets in oil–water systems [12]. Egg white proteins can be quickly adsorbed and directionally arranged at the oil–water interface to form a facial mask [13]. XG and CA anionic groups formed more electrostatic connections with egg white proteins, which interfered with the interactions between hydrophilic residues. This would affect the tertiary structure of the protein and adjust the amphiphilic balance, which could improve the flexibility of protein molecule adsorption and arrangement at the oil–water interface, and ultimately improve the EAI of the EWL + XG and EWL + CA groups [27].

Dispersive droplets are thermodynamically unstable and influenced by coalescence, flocculation, and creaming [28]. Coalescence is the fusion of oil droplets caused by the rupture of the interface film of close oil droplets. Droplets stabilized by protein–polysaccharide complexes in the EWL + XG and EWL + CA groups tend to have greater kinetic stability by preventing coalescence owing to the high intensity of the electrostatic repulsion force [28]. Therefore, the ESI of these groups was improved.

In addition, the degree of substitution of the pyruvate group in XG (30–40%) and the sulfate group in CA (25–30%) indicated that the electrostatic effect between XG and egg white protein was stronger, and there were many binding sites, which could form more polysaccharide–protein complexes and ultimately facilitate egg white protein adsorption on the interface and exert its emulsifying properties [29]. However, the formation of polysaccharide–protein complexes may not be conducive to the surface activity of egg white proteins at the oil–water interface because of the structure of the guar polymer chain [5], leading to a reduction in the emulsifying property.

#### 3.2.2. Effect of Polysaccharides on Adsorption of Interfacial Proteins in Emulsion

The effects of different polysaccharides on the adsorption capacity of the interfacial proteins in the emulsions are shown in Figure 4. Significant differences were observed between the experimental and control groups. Compared with the control group, the protein adsorption capacity at the oil–water interface in the EWL + XG and EWL + CA groups was significantly increased (*p* < 0.05), whereas the protein adsorption capacity in the EWL + GG group was significantly decreased (*p* < 0.05).

It was confirmed that XG and CA facilitated the adsorption of proteins at the oil–water interface [8]. Therefore, the EAI and ESI of the EWL + XG and EWL + CA groups improved. Flocculation of emulsions is caused by proteins unable to completely cover the droplet surface or the low interfacial film thickness, causing proteins to be adsorbed simultaneously at the surface of multiple droplets. To prevent oil droplet flocculation, the interfacial protein adsorption layer should reach a certain thickness [30]. The anionic polysaccharides, XG and CA, formed more cohesive complexes owing to the electrostatic attraction between them and the positively charged region of the protein, thereby increasing the boundary film thickness. The interfacial protein adsorption in the EWL + XG group had a higher value, with an optimal emulsifying effect. Chang et al. [31] found that the protein–pectin (negative) multilayer assembly aggregates improved emulsion stability by increasing the interface film thickness. The addition of GG leads to a decrease in interfacial protein adsorption at the oil–water interface, which directly leads to the inability of the protein to cover the droplets produced during homogenization, resulting in decreased EAI and ESI [32]. Studies have shown that the increased molecular flexibility of glycosylated proteins can promote the adsorption, swelling, and redistribution of proteins at the oil–water interface and ultimately improve the emulsification properties of proteins [33]. GG and egg white protein complexes form complexes through disulfide bonds and hydrophobic interactions, etc., which may make the flexibility of egg white protein poor and not conducive to its adsorption at the interface [34].

#### 3.2.3. Effect of Polysaccharides on the Creaming Index of Emulsions

As shown in Figure 5, the CI of the experimental and control groups gradually increased with time. From 0 to 5 h, the CI of the EWL, EWL + GG, EWL + CA, and EWL + XG groups increased from 0% to 2.60%, 2.64%, 2.30%, and 0.84%, respectively, indicating that the EWL + XG group had the best storage stability.

Among emulsion instability mechanisms, density–driven creaming is the most common phenomenon in emulsion storage and is inversely proportional to the viscosity of the liquid dispersed [31]. XG and CA improved the viscosity of the liquid to slow the continuous precipitation of the water phase in the emulsion with the thickening effect; therefore, the CI of the EWL + XG and EWL + CA groups decreased. Although GG also improved the viscosity of the liquid, its effect on the egg white protein structure limited the adsorption of proteins at the oil–water interface. Furthermore, unlike XG and CA, there was no charge repulsion effect on the surface of the droplets; thus, simply increasing the viscosity of the dispersion had no beneficial effect on CI.

### 3.3. Intermolecular Interaction of the Polysaccharide–Egg White Protein Complexes

#### 3.3.1. Hydrogen Bonds and Electrostatic Interactions between Polysaccharides and Egg White Protein

Figure 6 shows the variation trend of apparent viscosity with the shear rate of EWL or the polysaccharide–EWL composite, with and without NaCl or urea. The apparent viscosity of each sample decreased with increasing shear rate, indicating the behavior of shear–thinning pseudoplastic non–Newtonian fluids. This is mainly because entangled proteins and polysaccharides are oriented along the flow direction, and these molecules are straightened, oriented, and unwrapped, resulting in shear–thinning [35].

Urea and NaCl were selected as hydrogen bond–breaking and electrostatic–interfering agents to explore the effect of hydrogen bonding and electrostatic interactions on the apparent viscosity of the liquid, which could characterize the intramolecular or intermolecular interactions between polysaccharides and proteins [36,37]. With the addition of NaCl to the mixture, Cl^–^ competed with the negative polysaccharides for the binding sites of positively charged proteins, and Na^+^ competed with the positively charged proteins for the binding sites of the negative polysaccharides, which shielded the electrostatic attraction between the negative polysaccharides and proteins [20]. Because of their ability to form strong hydrogen bonds, urea functional groups preferentially adsorb onto protein hydrophilic residues through hydrogen bonds, reducing the intramolecular and intermolecular interactions of proteins and polysaccharides, which leads to decreased solution viscosity [38,39]. After shielding the electrostatic and hydrogen–bond interactions, the initial apparent viscosity decreased to varying degrees because the addition of NaCl or urea blocked the intramolecular and intermolecular interactions of polysaccharides and proteins, and the phenomenon of molecular entanglement was weakened, resulting in decreased apparent viscosity [40]. After shielding the specific interaction force, the decrease in apparent viscosity indicated that compared with the EWL group, the electrostatic interaction was stronger in the EWL + XG group (3.6 Pa·s vs. 0.8 Pa·s), whereas the hydrogen bond and electrostatic interaction were stronger in the EWL + CA group (3.0 Pa·s vs. 1.8 Pa·s, 1.7 Pa·s vs. 0.8 Pa·s). The change in apparent viscosity after the addition of GG was similar to that of the EWL group.

XG and CA are polyanions in aqueous solutions because their side chains contain acidic groups, which attract egg white proteins with positive electric regions and hinder the fluidity of the protein chains [40]. CA has eight hydrogen bond donors and 25 hydrogen–bond receptors [6]. Therefore, the electrostatic interaction between XG and egg white protein was stronger, and the hydrogen bonds and electrostatic interactions between CA and egg white protein were stronger. Compared with individual proteins, amphiphilic macromolecules formed by electrostatic recombination between oppositely charged proteins and polysaccharides may change the conformation of proteins, making them more favorably adsorbed or expanded [41] and better anchored on the oil–water interface [9]. Thus, the functional properties of the EWL + XG and EWL + CA groups were enhanced.

#### 3.3.2. Effects of Polysaccharides on the Hydrophobic Interactions between Egg White Proteins

Changes in surface hydrophobicity strongly affected the functional properties of proteins (such as solubility, interfacial characteristics, and emulsifying activity). A greater number of balanced hydrophobic groups distributed with the hydrophilic amino acids was exposed in proteins, which could improve the functional properties of the proteins [42]. As shown in Figure 7, compared with the control group, all three polysaccharides significantly reduced the surface hydrophobicity of egg white protein (*p* < 0.05). The reduction rates in the EWL + XG, EWL + CA and EWL + GG groups were 30.69%, 13.13%, and 16.34%, respectively.

Proteins are polymorphic polymers with high structural variability [27]. Duan et al. [43] showed that the interfacial spreading ability of proteins depended on the conformation, with a certain degree of denaturation conducive to exerting the functional characteristics of the protein. The three polysaccharides changed the egg white protein environment and induced it to expose more hydrophobic regions after denaturation, which resulted in insufficient Coulomb repulsion between molecules to offset long–distance hydrophobic attraction [44] and ultimately forced hydrophobic groups to aggregate under hydrophobic interactions. Therefore, the surface hydrophobicity was reduced [45]. Khan et al. [46] and Li et al. [24] also found that the interaction between proteins and polysaccharides caused changes in protein secondary structure and aggregation, and that hydrophobic amino acids were less exposed to the protein surface in the presence of polysaccharides. However, the mechanical force in the process of stirring and homogenization destroys and disperses the aggregated and denatured proteins, causing the hydrophobic groups to be fully exposed, allowing proteins to quickly undergo the stages of diffusion, adsorption, structural rearrangement, crosslinking, and solidification to form an interfacial film of bubbles or emulsion droplets [47]. In our results, the surface hydrophobicity of the EWL + XG group significantly decreased; therefore, the hydrophobic aggregation was stronger, and the mechanical dispersion effect was easier. Consequently, the functional characteristics of EWL were significantly improved.

#### 3.3.3. Effects of Polysaccharides on the Sulfhydryl Content of Egg White Protein

The surface sulfhydryl group is an important active group that reflects the structural changes in proteins and is closely related to their functional characteristics. XG significantly increased the surface sulfhydryl content of egg white protein by 35% (Figure 8) (*p* < 0.05), possibly because XG denatures more egg white protein and exposes more sulfhydryl groups. Wang et al. [48] found that the surface sulfhydryl content of rice bran protein hydrolysate increased with elevated ferulic acid and that its EAI and ESI were also improved.

The change in the free sulfhydryl content reflects the breaking and formation of protein intermolecular disulfide bonds, which are related to protein denaturation and aggregation [14]. The only protein containing a free sulfhydryl group in egg whites is ovalbumin; lysozyme and ovotransferrin contain 4 and 15 disulfide bonds, respectively [49]. The addition of GG significantly reduced the free sulfhydryl group content of egg white protein from 19.34 to 17.86 μmol/g. This finding could be attributed to the fact that GG addition induces aggregation of denatured egg white protein molecules into filamentous proteins, which is related to disulfide bonding [50]. The increase in disulfide bonds led to a decrease in free sulfhydryl groups. Moreover, the molecular flexibility of the newly formed aggregates was poor, which is not conducive to forming an interfacial film [10]; thus, the FA, ESI, and EAI of the EWL + GG group were all reduced.

### 3.4. Effect of Polysaccharides on the Microstructure of Egg White Protein

The aggregation of protein–polysaccharide complexes significantly affects the functional characteristics of the protein so that it can be an effective emulsifier or foaming agent [51]. In Figure 9a, the green fluorescent area indicates egg white protein, and the black area indicates polysaccharides or water. In the absence of polysaccharides, egg white protein showed slight aggregation, possibly because of the electrostatic interaction between negatively charged ovalbumin and positively charged lysozyme under a pH of 8.2 [52]. With the addition of the three polysaccharides, protein aggregation increased, and the states of irregular and filamentous aggregates were observed. The EWL + CA and EWL + XG groups showed irregular aggregates, whereas the EWL + GG group showed filamentous aggregates.

Polysaccharides can be adsorbed to proteins through noncovalent interactions and cause conformational changes or unfolding of egg white protein, leading to the exposure of sequences with an aggregation tendency in the protein to bridge several protein molecules, further aggregating into larger water–soluble protein–polysaccharide complexes (Figure 9b). XG and CA assemble molecules, leading to complex aggregation; the aggregation state of the EWL + XG group is more conducive to the functional characteristics of the EWL. Mohsen and Samira [7] also found that increased spherical agglomerates enhanced protein adsorption at the interface, thus improving the emulsifying and foaming abilities. Filamentous aggregates are small oligomers or long fibrous structures of several to 100 nm formed between the side chains of denatured proteins or polypeptide chains through complex mechanisms [53]. Moderately aggregated proteins with long fibrous structures are stacked to form fibrin and filamentous aggregates [53]. This aggregation state is very rigid and not conducive to adsorption, expansion, and rearrangement at the interface [10]. Therefore, this limits the exertion of the functional characteristics of egg white proteins in the EWL + GG group.

## 4. Conclusions

The association between molecular interactions, aggregation states, and functional properties of polysaccharide–protein complexes was systematically investigated for the first time. Compared with adding GG and CA, adding XG has a better effect on improving the foaming and emulsifying properties of egg white liquid. The polysaccharide–egg white complex is aggregated through intermolecular interaction, which has different effects on the foaming and emulsifying properties of egg white protein. This study provides a theoretical basis and technical support for developing and utilizing special EWLs to meet the market demands of baking and other industries with strong functionality. In the future, we can develop a highly functional egg white liquid by adding polysaccharides, which can save the use cost of egg white liquid and increase the characteristics of the product.

## Figures and Tables

**Figure 1 foods-11-02119-f001:**
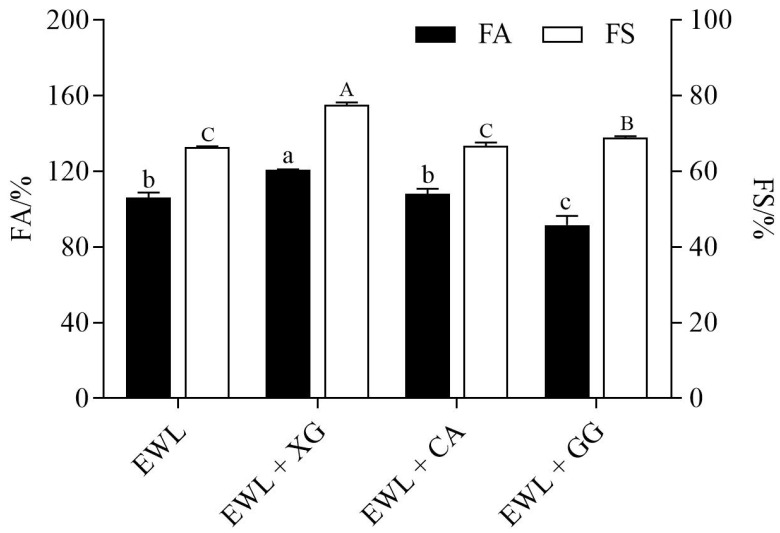
Effects of polysaccharide added to egg white liquid on FA and FS. Different lowercase letters indicated significant differences (*p* < 0.05) between FA groups, and different uppercase letters indicated significant differences (*p* < 0.05) between FS groups.

**Figure 2 foods-11-02119-f002:**
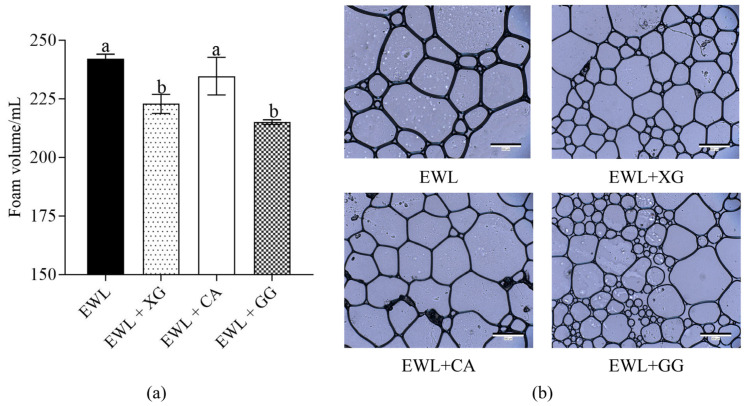
Effects of polysaccharide added to egg white liquid on foam structure. (**a**) The foam volume after full foaming in the presence of polysaccharides of egg white liquid. Different letters indicate a significant difference (*p* < 0.05) between groups. (**b**) The microscopic images of the foam in the presence of polysaccharides of egg white liquid (10×).

**Figure 3 foods-11-02119-f003:**
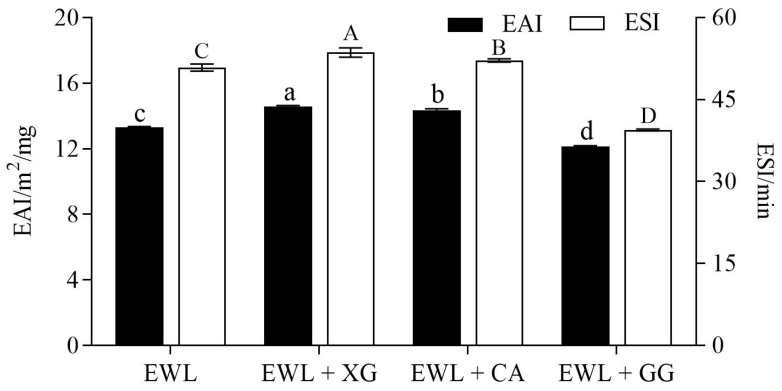
Effects of polysaccharide added to egg white liquid on EAI and ESI. Different lowercase letters indicate a significant difference (*p* < 0.05) between EAI groups, and different uppercase letters indicate a significant difference (*p* < 0.05) between ESI groups.

**Figure 4 foods-11-02119-f004:**
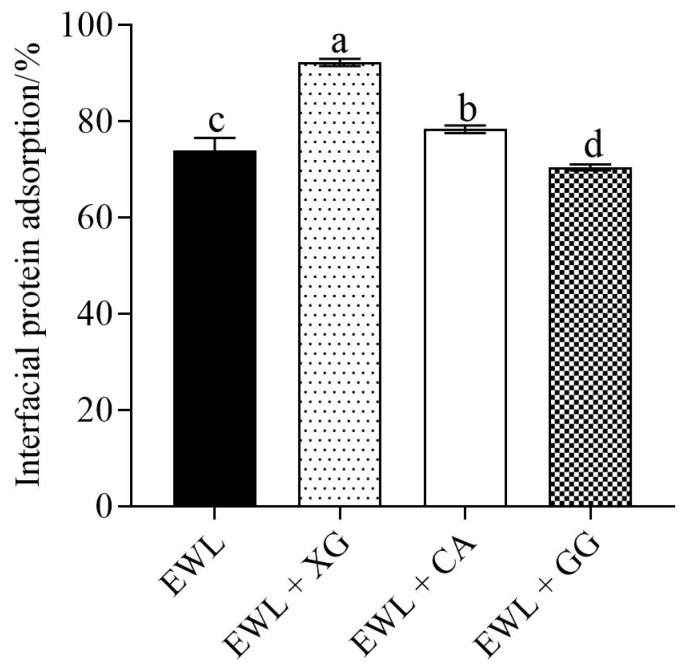
Effects of polysaccharide added to egg white liquid on the interfacial protein adsorption at oil–water. Different letters indicate a significant difference (*p* < 0.05) between groups.

**Figure 5 foods-11-02119-f005:**
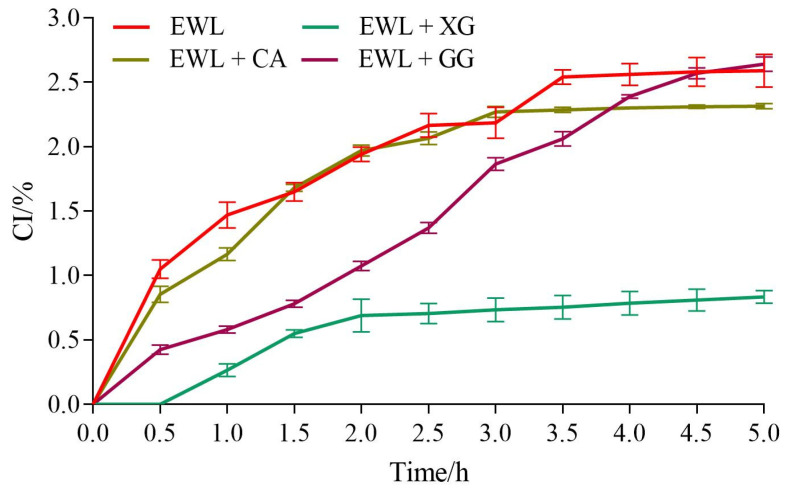
The effects of polysaccharide added to egg white liquid on CI.

**Figure 6 foods-11-02119-f006:**
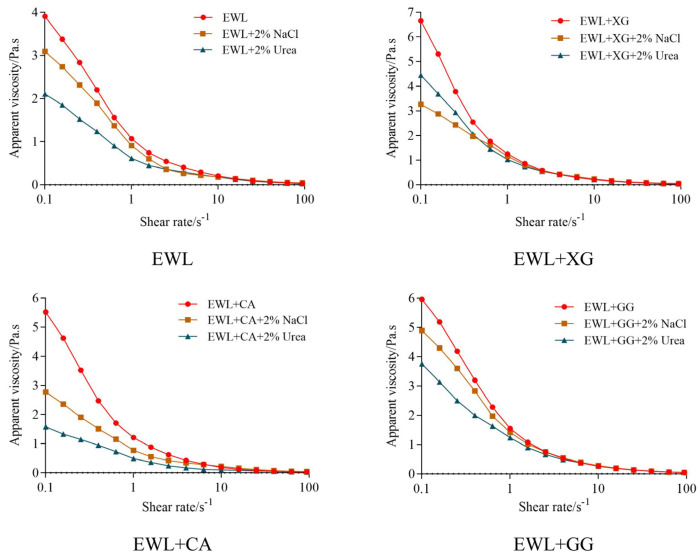
Electrostatic interaction and hydrogen bonding between polysaccharides and egg white protein.

**Figure 7 foods-11-02119-f007:**
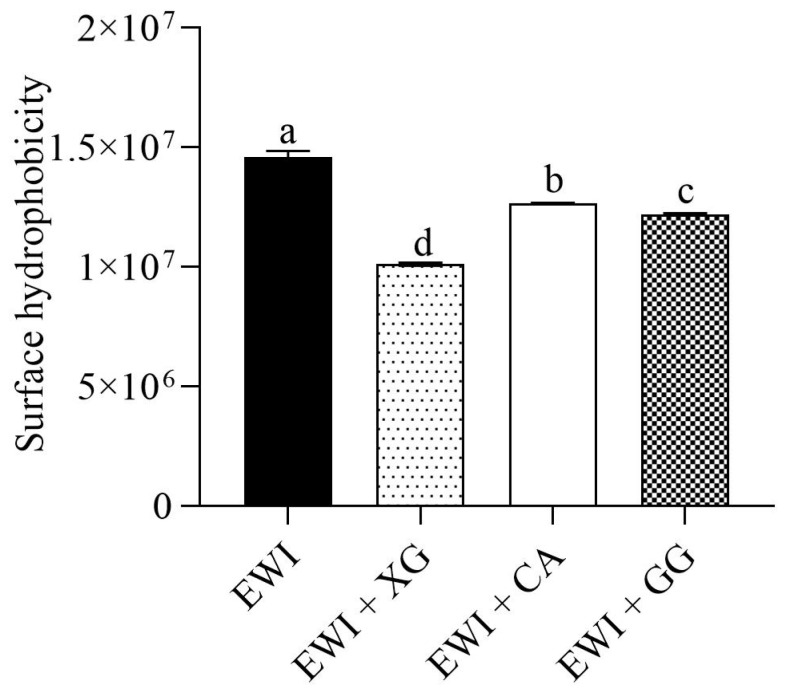
Effects of polysaccharide added to egg white liquid on surface hydrophobicity. Different letters indicate significant differences (*p* < 0.05) between groups.

**Figure 8 foods-11-02119-f008:**
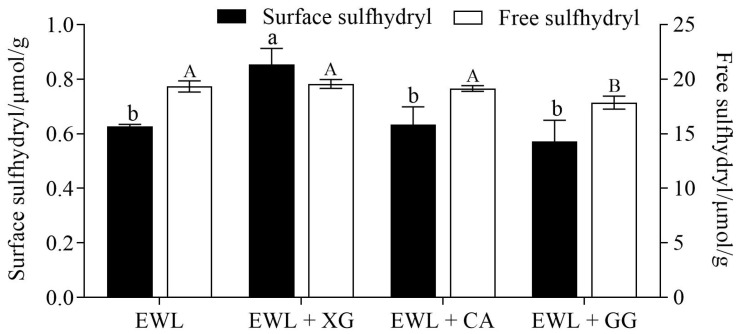
Effects of polysaccharide added to egg white liquid on sulfhydryl content. Different lowercase letters show significant differences (*p* < 0.05) in surface sulfhydryl of egg white protein between groups, and different uppercase letters show significant differences (*p* < 0.05) in the free sulfhydryl of egg white protein between groups.

**Figure 9 foods-11-02119-f009:**
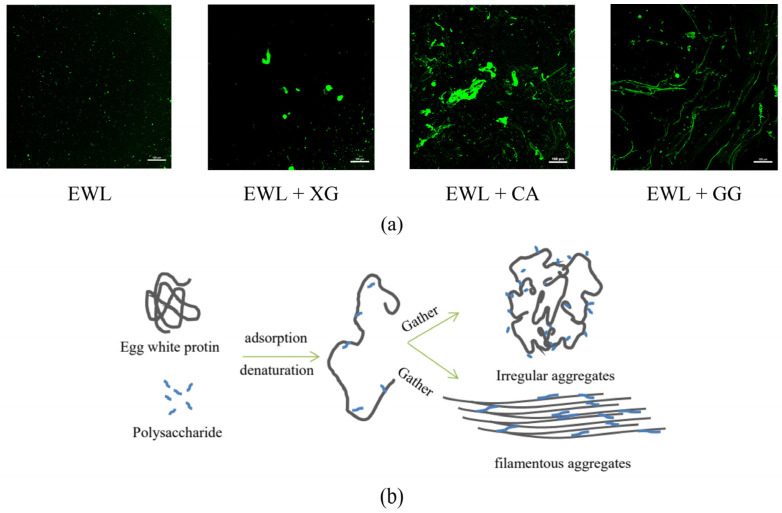
(**a**) Effects of polysaccharide added to egg white liquid on microscopic images. (**b**) Electrostatic interaction and hydrogen bonding between polysaccharides and egg white protein.

**Table 1 foods-11-02119-t001:** Effects of polysaccharide added to egg white liquid on average area and quantity of foam. Different letters in the same row indicate a significant difference (*p* < 0.05) between groups.

Items	EWL	EWL + XG	EWL + CA	EWL + GG
Number of foam	22	57	49	169
Average area of foam/mm^2^	0.127 ± 0.141 ^a^	0.052 ± 0.040 ^b^	0.120 ± 0.144 ^a^	0.022 ± 0.062 ^b^

Date expressed by mean ± standard deviation, and different letters in the same row indicate a significant difference (*p* < 0.05) between groups.

## Data Availability

The data that support the findings of this study are available upon request from the corresponding author. The data are not publicly available due to privacy or ethical restrictions.

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
