# Peer review of "Effect of Xanthan Gum, Kappa–Carrageenan, and Guar Gum on the Functional Characteristics of Egg White Liquid and Intermolecular Interaction Mechanism"

_foods, 2022, doi:10.3390/foods11142119_

Round 1
Reviewer 1 Report
The authors presented the effect of some commercial hydrocolloids on the functional characteristics of egg white liquid and intermolecular interaction mechanism. The idea of the article is good. Well designed. Appropriate analyses are measured and good discussions are provided. However, a few points to improve the current format of the article will be mentioned below:
The title could be modified to mention SOME COMMERCIAL HYDROCOLLOIDS.
The abstract should be more informative by giving real results rather than elastic sentences. Important and main contents should be given. Support the results with some quantitative data. Moreover, no conclusions are provided.
Minor editing of English language and style required. For example in line 37: The emulsifying properties is -> are; Line 78: of -> for; line 143: Remove the dot from the middle of the sentence; etc.
Line 70-76: WHAT IS THIS? IT IS WORRYING THAT IT HAS NOT BEEN READ ONCE SO THAT SUCH FLAWS ARE NOT OBSERVED.
Line 83: On what basis was the concentration of 0.2% of the hydrocolloids selected? Do they have the same viscosity at this concentration? What will be the basis of comparison?
The analysis you provide in line 213 should also include the Guar gum. Because this gum is also anionic, therefore, it is necessary to provide a correct discussion and clearly state the reason for the difference in the behavior of guar gum with the other two hydrocolloids in the formation and stability of the egg white foam.
Line 250: …. the foam size smaller and denser. Be sure to calculate the size of the bubbles and their size distribution and uniformity with the image processing method and add them to the results.
Line 326: Why does the addition of GG reduce the interfacial protein adsorption at the oil-water interface? Give strong and documented reasons.
Lines 338-345: Why was CI of EWL-XG higher than EWL-GG? It is inconsistent with previous observations and results. Give clear reasons and possibilities for this phenomenon.
Line 440: Why the addition of CA did not reduce the free sulfhydryl group content of egg white protein?
Conclusion: what is the future of your findings? Conclusion is not insightful, what are suggestions?
Author Response
Dear reviewers:
Thank you for your comments concerning our manuscript entitled “Effect of xanthan gum, kappa-carrageenan, and guar gum on the functional characteristics of egg white liquid and intermolecular interaction mechanism” (ID: foods-1807438). Those comments are all valuable and very helpful for revising and improving our paper, as well as the important guiding significance to our researcher. We have studied comments carefully and have made a correction which we hope meet with approval. Revised portions are marked up using the “Track Changes” on the manuscript. The main correction in this manuscript and the responses to reviewer’s comments are below:
Response to Reviewer 1 Comments:
-------------------------------------------------------------------
Major comments:
Comment #1:
The title could be modified to mention some commercial hydrocolloids.
Response 1:
Thank you for your advice. We have revised the title to “Effect of xanthan gum, kappa-carrageenan, and guar gum on the functional characteristics of egg white liquid and intermolecular interaction mechanism”. Please see line 2-4.
Comment #2:
The abstract should be more informative by giving real results rather than elastic sentences. Important and main contents should be given. Support the results with some quantitative data. Moreover, no conclusions are provided.
Response 2:
Thank you for your advice. We modified the sentence expression of the abstract. Data and important contents are added, and the main conclusions are supplemented. Please see line 12-29.
Comment #3:
Minor editing of English language and style required. For example in line 37: The emulsifying properties is -> are; Line 78: of -> for; line 143: Remove the dot from the middle of the sentence; etc.
Response 3:
Thank you for your advice. We already revised these issues. Please see line 42, 77, 143.
Comment #4:
Line 70-76: What is this? It is worrying that it has not been read once so that such flaws are not observed.
Response 4:
Thank you for your correction. We have deleted this section.
Comment #5:
Line 83: On what basis was the concentration of 0.2% of the hydrocolloids selected? Do they have the same viscosity at this concentration? What will be the basis of comparison?
Response 5:
The concentration range of polysaccharide selected in the previous related test was 0.15~0.20%. On this basis, we carried out a multi-concentration gradient experiment. The results showed that only when the concentration of added polysaccharide was less than or equal to 0.2%, can the three polysaccharide-egg white complex liquid be close to the original state of egg white liquid, so the final concentration was 0.2%. In this experiment, the purpose of our experimental design was to compare the effect of different polysaccharides on the functional properties of egg white liquid at the same dosage. Therefore, we did not consider the difference in polysaccharide viscosity at the concentration of 0.2%.
Comment #6:
The analysis you provide in line 213 should also include the Guar gum. Because this gum is also anionic, therefore, it is necessary to provide a correct discussion and clearly state the reason for the difference in the behavior of guar gum with the other two hydrocolloids in the formation and stability of the egg white foam.
Response 6:
The data we reviewed showed that guar gum is a neutral polysaccharide, so we discussed guar gum separately. Please see line 224-228.
The relevant references are as follows:
- Ganie, S. A., et al. "Preparation, characterization, release and antianemic studies of guar gum functionalized Iron complexes." International Journal of Biological Macromolecules 183(2021):1495-1504.
- Osamu, et al. "Effect of a Phosphorylated Guar Gum Hydrolysate on Increased Calcium Solubilization and the Promotion of Calcium Absorption in Rats." Bioscience, Biotechnology, and Biochemistry 64.1(2014):160-166.
Comment #7:
Line 250: …. the foam size smaller and denser. Be sure to calculate the size of the bubbles and their size distribution and uniformity with the image processing method and add them to the results.
Response 7:
Thank you for your advice. In Figure 2(b), we calculated the number of foam and average foam area of each picture to determine size distribution and uniformity of foam, and provide data support for our results. Please see line 251-252, 261-264.
Comment #8:
Line 326: Why does the addition of GG reduce the interfacial protein adsorption at the oil-water interface? Give strong and documented reasons.
Response 8:
Thank you for your advice. We have provided additional discussions and corresponding references in the text. Please see line 336-341.
Comment #9:
Lines 338-345: Why was CI of EWL-XG higher than EWL-GG? It is inconsistent with previous observations and results. Give clear reasons and possibilities for this phenomenon.
Response 9:
Thank you for your advice. CI of EWL+XG group was lowest compared to EWL, EWL+GG, and EWL+CA, so the EWL+XG group had the best storage stability. This is consistent with previous results of EAI, ESI, and interfacial protein adsorption. Plese see line 343-346.
Comment #10:
Line 440: Why the addition of CA did not reduce the free sulfhydryl group content of egg white protein?
Response 10:
The decrease of free sulfhydryl group indicates that the molecules in the egg white liquid form disulfide bonds. CA did not reduce the content of free sulfhydryl in egg white protein, indicating that disulfide bond was not the main force in EWI+CA group. Xanthan gum, which is also a negative polysaccharide, did not reduce the content of free sulfhydryl. In GG group, the content of free sulfhydryl group decreased significantly, and the formation of disulfide bond reduced the flexibility of the molecule, which was not conducive to the adsorption of proteins at the interface.
Comment #11:
Conclusion: what is the future of your findings? Conclusion is not insightful, what are suggestions?
Response 11:
Done. Please see line 490-500.
Comment:
Moderate English changes required.
Response:
Thanks. We have carefully edited the entire manuscript and the manuscript has been polished by a professional company. The proof of English polishing was attached.

Reviewer 2 Report
Manuscript foods-1807438, entitled “Effect of polysaccharides on the functional characteristics of egg white liquid and intermolecular interaction mechanism”
The article provides useful information about the effects of polysaccharides on the functional characteristics of egg white liquid and intermolecular interaction mechanism. Although, the experiment is in general appropriately designed and implemented, some points should be corrected or clarified.
General comments
1. Please explain abbreviations, when they are initially used in text
2. How many samples per group for each measurement?
3. In some parts, authors reach to incorrect conclusions. For example:
a. L221: Only in EWL+XG
b. L239-240: Not for EWL+CA
c. L249: Three or two?
d. L250-252: Compared to? Please check L261-262
e. L482-484: Also for GG?
Minor points:
L37: “…properties are also affected by the…”
L47-48: “…not only possess nutritional and health properties but also provide improved functional characteristics…”
L58: “…is difficult to be controlled and…”
L59: “could provide a solution in this problem” instead of “is a good research direction”
L64: “The objective of the present work….”
L70-76: Please delete
L161-162: “…between the two wavelengths was recorded…”
L173-174: Please rephrase
L203: “…control group (EWL), FA and FS of the…”
L214, 223: “due to” instead of “because of”
L227: “As time passes” instead of “With the extension of time”
L233: “…inclined to be adsorbed to the…”
L243: “attributed to” instead of “provided by”
L246-247: “…control (EWL) group, the foam volume of the EWL+XG and EWL+GG…”
L282-284: “It is necessary to use a large amount of mechanical energy provided by the homogenization process to increase the interface area and to disperse oil droplets in oil-water systems [12].”
L309-311: “…groups was significantly increased (p<0.05), whereas the protein adsorption capacity in the EWL+GG group was significantly decreased (p<0.05).”
L318: “…causing proteins to be adsorbed…”
L323-324: “The interfacial protein adsorption in the EWL+XG group has higher value, with an optimal…”
L374-378: Where are these results shown?
L382-384: Please rephrase
L419: “…group significantly decreased; therefore…”
L440: “This finding could be attributed to the fact that GG addition” instead of “This is because adding GG”
L448: What do you mean by “which should make it”?
L460: “Polysaccharides can be adsorbed to proteins…”
Author Response
Dear reviewers:
Thank you for your comments concerning our manuscript entitled “Effect of xanthan gum, kappa-carrageenan, and guar gum on the functional characteristics of egg white liquid and intermolecular interaction mechanism” (ID: foods-1807438). Those comments are all valuable and very helpful for revising and improving our paper, as well as the important guiding significance to our researcher. We have studied comments carefully and have made a correction which we hope meet with approval. Revised portions are marked up using the “Track Changes” on the manuscript. The main correction in this manuscript and the responses to reviewer’s comments are below:
Response to Reviewer 2 Comments:
-------------------------------------------------------------------
Major comments:
General comments
Comment #1:
Please explain abbreviations, when they are initially used in text.
Response 1:
Thank you for your advice. We added explanations for abbreviations that are initially used. Please see line 88, 104, and 125.
Comment #2:
How many samples per group for each measurement?
Response 2:
Experiments of each group were conducted in triplicate. We added this explanation in line 196.
Comment #3:
In some parts, authors reach to incorrect conclusions. For example:
L221: Only in EWL+XG
Response 3:
Done. Please see line 224.
Comment #4:
L239-240: Not for EWL+CA
Response 4:
Done. Please see line 239, 241.
Comment #5:
L249: Three or two?
Response 5:
Done. Please see line 252-254.
Comment #6:
L250-252: Compared to? Please check L261-262.
Response 6:
Thank you for your comments. We supplemented data description reflecting differences in foam size between groups to support our statement. Please see line 254-255, 268-269.
Comment #7:
L482-484: Also for GG?
Response 7:
Done. This part of the content has been revised. Please see line 490-500.
Minor points:
Comment #8:
L37: “…properties are also affected by the…”
Response 8:
Done. Please see line 42.
Comment #9:
L47-48: “…not only possess nutritional and health properties but also provide improved functional characteristics…”
Response 9:
Done. Please see line 52-53.
Comment #10:
L58: “…is difficult to be controlled and…”
Response 10:
Done. Please see line 63-64.
Comment #11:
L59: “could provide a solution in this problem” instead of “is a good research direction”
Response 11:
Done. Please see line 65.
Comment #12:
L64: “The objective of the present work….”
Response 12:
Done. Please see line 70.
Comment #13:
L70-76: Please delete
Response 13:
Done. We have deleted this sentence.
Comment #14:
L161-162: “…between the two wavelengths was recorded…”
Response 14:
Done. Please see line 161-162.
Comment #15:
L173-174: Please rephrase
Response 15:
Done. Please see line 172-173.
Comment #16:
L203: “…control group (EWL), FA and FS of the…”
Response 16:
Done. Please see line 206.
Comment #17:
L214, 223: “due to” instead of “because of”
Response 17:
Done. Please see line 225.
Comment #18:
L227: “As time passes” instead of “With the extension of time”
Response 18:
Done. Please see line 229.
Comment #19:
L233: “…inclined to be adsorbed to the…”
Response 19:
Done. Please see line 235.
Comment #20:
L243: “attributed to” instead of “provided by”
Response 20:
Done. Please see line 245.
Comment #21:
L246-247: “…control (EWL) group, the foam volume of the EWL+XG and EWL+GG…”
Response 21:
Done. Please see line 248-249.
Comment #22:
L282-284: “It is necessary to use a large amount of mechanical energy provided by the homogenization process to increase the interface area and to disperse oil droplets in oil-water systems [12].”
Response 22:
Done. Please see line 289-291.
Comment #23:
L309-311: “…groups was significantly increased (p<0.05), whereas the protein adsorption capacity in the EWL+GG group was significantly decreased (p<0.05).”
Response 23:
Done. Please see line 317-318.
Comment #24:
L318: “…causing proteins to be adsorbed…”
Response 24:
Done. Please see line 325.
Comment #25:
L323-324: “The interfacial protein adsorption in the EWL+XG group has higher value, with an optimal…”
Response 25:
Done. Please see line 330-331.
Comment #26:
L374-378: Where are these results shown?
Response 26:
Thank you for your advice. We added data to illustrate our results. In addition, we consider that the difference of the apparent viscosity between the group with urea or NaCl and that of without urea or NaCl could reflect the shielding effect of these two shielding agents for specific interactions. Comparing these differences between the EWL group and the polysaccharide-added group could reflect the strength of the specific intermolecular interaction force, so as to determine the main intermolecular force of experimental group. Please see line 387-392.
Comment #27:
L382-384: Please rephrase.
Response 27:
Done. Please see line 396-398.
Comment #28:
L419: “…group significantly decreased; therefore…”
Response 28:
Done. Please see line 433.
Comment #29:
L440: “This finding could be attributed to the fact that GG addition” instead of “This is because adding GG”
Response 29:
Done. Please see line 454-455.
Comment #30:
L448: What do you mean by “which should make it”?
Response 30:
We have modified this statement. Please see line 462.
Comment #31:
L460: “Polysaccharides can be adsorbed to proteins…”
Response 31:
Done. Please see line 474.

Round 2
Reviewer 1 Report
The authors have revised most of what has been said.